# UnModNet: Learning to Unwrap a Modulo Image for High Dynamic Range Imaging

**Chu Zhou**[1] **Hang Zhao**[2] **Jin Han**[1] **Chang Xu**[3] **Chao Xu**[1] **Tiejun Huang**[4,5] **Boxin Shi**[4,5*]

[1]Key Lab of Machine Perception (MOE), Dept. of Machine Intelligence, Peking University
[2]Massachusetts Institute of Technology
[3]School of Computer Science, Faculty of Engineering, University of Sydney
[4]NELVT, Dept. of Computer Science, Peking University
[5]Institute for Artificial Intelligence, Peking University
{zhou_chu, hanjin1619, tjhuang, shiboxin}@pku.edu.cn,
zhaohang0124@gmail.com, c.xu@sydney.edu.au, xuchao@cis.pku.edu.cn

## Abstract

A conventional camera often suffers from over- or under-exposure when recording a real-world scene with a very high dynamic range (HDR). In contrast, a modulo camera with a Markov random field (MRF) based unwrapping algorithm can theoretically accomplish unbounded dynamic range but shows degenerate performances when there are modulus-intensity ambiguity, strong local contrast, and color misalignment. In this paper, we reformulate the modulo image unwrapping problem into a series of binary labeling problems and propose a modulo edge-aware model, named as UnModNet, to iteratively estimate the binary rollover masks of the modulo image for unwrapping. Experimental results show that our approach can generate 12-bit HDR images from 8-bit modulo images reliably, and runs much faster than the previous MRF-based algorithm thanks to the GPU acceleration.

## 1 Introduction

Real-world scenes have a very high dynamic range (HDR) so that object contours are mostly lost in the over-exposed and under-exposed regions when captured by a conventional camera with a limited dynamic range and saved as an 8-bit image. To increase the dynamic range of captured images, many HDR reconstruction approaches have been proposed to increase the camera bit depth via hardware modifications [22, 36], as well as using computational methods to merge multi-bracketed captures [5] or a series of bursts [17]. Yet the dynamic range they can achieve is limited and the details of the HDR content often cannot be faithfully recovered. A modulo camera [59] can theoretically achieve unbounded dynamic range by recording the least significant bits of the irradiance signal, *i.e.*, the camera hardware "resets" the scene radiance arriving at the sensor before reading it out whenever it reaches saturation (*e.g.*, for an 8-bit image, 256 will be reset to 0 and re-start the counting again as long as the shutter keeps open). By unwrapping the captured modulo image with a customized Markov random field (MRF) based algorithm, the HDR image could be practically restored, as shown in Figure 1 (top left). We denote the irradiance of an HDR image as $\mathbf{I} = \{I(x, y, c)\}$, and its corresponding modulo image as $\mathbf{I}_m = \{I_m(x, y, c)\}$, where $(x, y)$ is the pixel coordinate and $c$ denotes the color channel index. $\mathbf{I}_m$ is equivalent to the least significant $N$ bits of $\mathbf{I}$. As illustrated in the bottom row of Figure 1, their relationship can be expressed as:

$$\mathbf{I}_m = \texttt{mod}(\mathbf{I}, 2^N) \quad \text{or} \quad \mathbf{I} = \mathbf{I}_m + 2^N \cdot \mathbf{K}, \tag{1}$$

where $\mathbf{K} = \{K(x, y, c)\}$ is the number of *rollovers* per pixel.

---

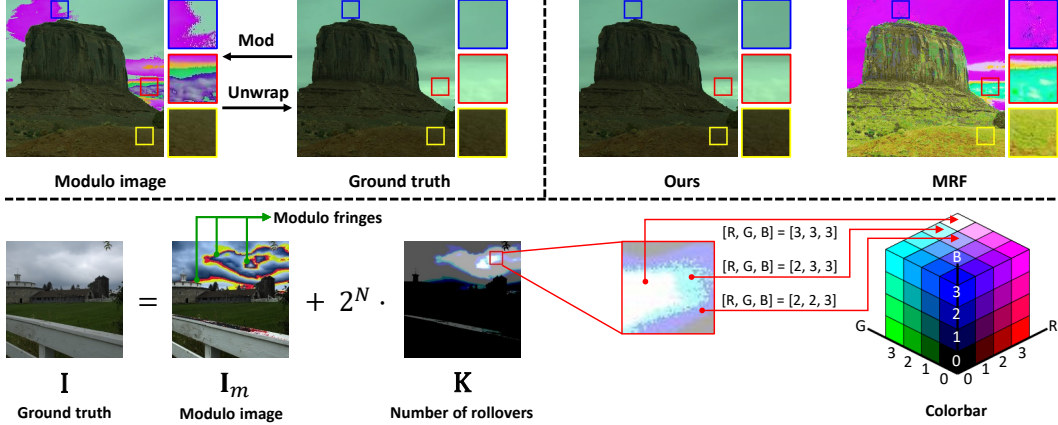

Figure 1: Top left: a modulo camera mods ("resets") the scene radiance and turns it into a modulo image, and the HDR reconstruction algorithm unwraps the modulo image back to the ground truth. Top right: fundamental problems of the MRF-based algorithm [59] and how the proposed UnModNet solves these issues; blue, red, and yellow boxes highlight examples for **modulus-intensity ambiguity**, **strong local contrast**, and **color misalignment**, respectively. Bottom row: the relationship between the ground truth HDR image **I**, the modulo image $\mathbf{I}_m$ (examples of modulo fringes are marked with green arrows), and the number of rollovers **K** (linearly scaled to $[0, 255]$ for visualization) in Equation (1).

However, as shown in Figure 1 (top right), the MRF-based unwrapping algorithm [59] is not robust due to several fundamental issues:

(1) **Modulus-intensity ambiguity.** It is difficult to discriminate whether a pixel value is a modulus or an intensity (non-modulus). The previous method would often incorrectly unwrap non-modulo pixels as it used a cost function without a data term during optimization.

(2) **Strong local contrast.** Dense *modulo fringes* (marked with green arrows in the bottom row of Figure 1) are usually caused by strong local contrast in irradiance. The previous method often fails around these regions as it only focused on local smoothness and ignored contextual information and structural patterns.

(3) **Color misalignment.** The previous method independently unwraps each color channel, resulting in severe color misalignment artifacts across three channels, so it cannot handle RGB images robustly.

In this paper, we reformulate the unwrapping of a modulo image into a series of binary labeling problems and propose a learning-based framework named *UnModNet*, as shown in Figure 2, to iteratively estimate the binary rollover mask of the input modulo image. Concretely, we have some key observations on the characteristics of modulo images: continuous irradiance regions are split up by the modulo operation, resulting in a large edge magnitude around modulo fringes; the over-exposed regions are concentratedly distributed in an image, which makes the modulo pixels likely to cluster in local regions. Based on these unique features of modulo pixels and edges, we design UnModNet to be two-stage accordingly: the first stage is a modulo edge separator that estimates channel-wise edges unique to modulo images; the second stage is a rollover mask predictor that achieves high-accuracy rollover mask prediction with the guidance of modulo edges.

To summarize, our learning strategy for modulo image unwrapping proposes three customized model designs to solve the three issues in the previous MRF-based algorithm [59] as follows:

(1) **Modulo edge separator** is proposed to distinguish the semantic and boundary information of the scene to relieve the modulus-intensity ambiguity and indicate correct regions to unwrap in a context-aware manner.

(2) **Rollover mask predictor** is adopted to deal with strong local contrast and dense modulo fringes to increase the capability of unwrapping a higher dynamic range in a structure-aware manner.

(3) **Consistent color prediction** is achieved by joint unwrapping across RGB channels, so that our model restores natural color appearance reliably.

Experimental results show that our approach can generate 12-bit HDR images from 8-bit modulo images reliably, and runs much faster than the previous MRF-based algorithm [59] thanks to the GPU acceleration.

## 2 Related Work

**Multi-image HDR reconstruction.** One of the most representative multi-image HDR reconstruction methods, proposed by Debevec and Malik [5], merges several low dynamic range (LDR) photographs under different exposures. However, it suffers from ghosting artifacts in HDR results when there is misalignment caused by camera movement or scene change during the exposure time. This problem provokes a series of studies on ghosting removal in HDR images [25, 39, 43]. Instead of using bracketed exposures, Hasinoff *et al*. [17] fused a burst of frames of constant exposure, which reduces the exposure time substantially and makes alignment more robust. Recently, several deep convolutional neural networks (CNNs) based approaches [24, 56, 58] have been developed to rebuild an HDR image from multiple LDR images. In contrast, we focus on single-image HDR reconstruction.

**Single-image HDR reconstruction.** Single-image HDR reconstruction, which aims to reconstruct the HDR image from a single LDR image, is also named as inverse tone mapping [1]. It is free of ghosting artifacts but more challenging than its multi-image counterpart due to the lack of irradiance information in badly-exposed areas. This ill-posed problem can be solved by several approaches [32, 41] based on numerical optimization. Recently, two categories of new methods emerged: learning-based HDR restoration, which hallucinates plausible HDR content from a single LDR image; and unconventional cameras, which captures additional information in a single photo from the scene. Learning-based methods discover HDR image priors from a large amount of training data. HDRCNN [6] adopted an encoder-decoder architecture to restore saturated areas in LDR images. ExpandNet [31] concatenated and fused different levels of features extracted by CNN to get HDR images directly. Endo *et al*. [7] used CNN to predict the LDR images under multiple exposures and merged them by the classical method [5]. Metzler *et al*. [34] jointly optimized a diffractive optical element-based encoder and a CNN-based decoder to recover saturated scene details. Liu *et al*. [30] trained a CNN to reverse the camera pipeline to reconstruct the HDR image. Methods using unconventional cameras attempt to gather additional cues about dynamic range from the scene to address the ill-posed nature of this problem. Nayar *et al*. [36] placed an optical mask adjacent to a conventional image detector array to make spatially varying pixel exposures. Hirakawa and Simon [22] placed a combination of photographic filter over the lens and color filter array on a conventional camera sensor. Neuromorphic cameras are also shown to be useful in guiding the process of HDR imaging [16, 53, 55]. Furthermore, some concept cameras have been proposed. Tumblin *et al*. [49] proposed a log-gradient camera that does well in capturing detailed high contrast scenes. Zhao *et al*. [59] proposed a modulo camera-based framework to push out the boundary of the dynamic range.

**Phase unwrapping.** Phase unwrapping is a classic signal processing problem that refers to recovering the original phase value from the principal value (wrapped phase). It is widely used in domains like optical metrology [8], synthetic aperture radar (SAR) interferometry [15], and medical imaging [4]. Phase unwrapping can be solved by Poisson's equation [46], MRF-based iterative method [2], path-following method [20], *etc*. Recently deep CNNs have also been used to handle this problem [42, 47, 52]. However, these methods are designed for handling phase images, which have completely different properties from natural images.

**Natural image unwrapping.** Natural image unwrapping aims to recover the original scene radiance from its modulo counterpart, which is previously defined as what the modulo camera-based framework [59] tries to achieve. Although it is analogous to phase unwrapping, methods designed for phase unwrapping problem cannot be directly applied because phase images and natural images are two types of data with a huge domain gap. Recently, several solutions [28, 44, 45] have been proposed to deal with the natural image unwrapping problem. However, these methods require multiple modulo images as input and refuse to work when only a single modulo image is available. By exploring natural image statistics, the MRF-based algorithm proposed in [59] successfully demonstrates the

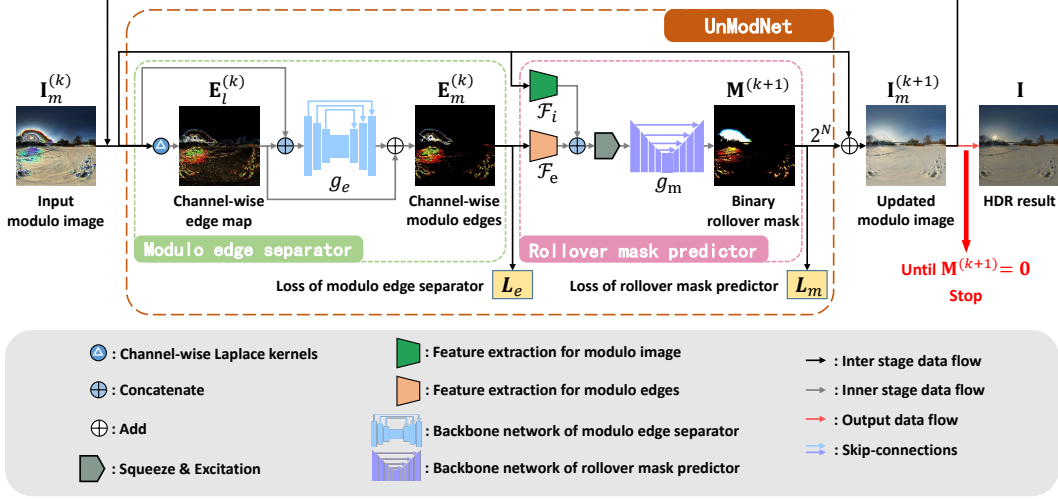

Figure 2: Architecture of the proposed UnModNet, which adopts a recursive structure to unwrap the modulo image step-by-step by estimating binary rollover masks iteratively. It consists of two stages: a modulo edge separator that separates channel-wise modulo edges, and a rollover mask predictor that predicts the binary rollover mask of the input modulo image with the guidance of modulo edges. The binary rollover mask $\mathbf{M}^{(k+1)}$ in the figure is multiplied by 255 for visualization.

feasibility of unwrapping a single modulo image to expand the dynamic range. However, failure cases are also commonly observed, as the example shown in Figure 1 (top right).

## 3 Method

In this section, we first introduce the iterative formulation of the problem, and show the overall pipeline of UnModNet in Section 3.1 and Figure 2. Then, we detail our two-stage UnModNet model designs in Section 3.2 and Section 3.3. Implementation details are presented in Section 3.4.

### 3.1 Problem formulation and overall pipeline

We aim to restore the ground truth HDR image $\mathbf{I}$ by unwrapping a single modulo image $\mathbf{I}_m$ captured by a modulo camera. According to Equation (1), it is equivalent to estimating the number of rollovers $\mathbf{K}$. Putting it into a probabilistic framework, our goal is to estimate $\mathrm{argmax}_{\mathbf{K}} P(\mathbf{K}|\mathbf{I}_m)$. Theoretically, the label space of $\mathbf{K}$ is the whole non-negative integer space $\{0, 1, 2, \ldots\}$. Given a modulo image, it is non-trivial to predict the label space either, which poses a major challenge to directly estimate the likelihood.

Therefore, we make the model more tractable by factorizing over the number of rollovers $\mathbf{K}$ as

$$P(\mathbf{K}|\mathbf{I}_m) = \prod_{k=1}^{\infty} P(\mathbf{M}^{(k+1)}|\mathbf{M}^{(1)}, \ldots, \mathbf{M}^{(k)}, \mathbf{I}_m) P(\mathbf{M}^{(1)}|\mathbf{I}_m), \tag{2}$$

where $\mathbf{M}^{(k)} = \{M^{(k)}(x, y, c)\}$ represents a binary rollover mask in $k$-th factor term which satisfies

$$M^{(k)}(x, y, c) = \begin{cases} 1 & \text{if } k \leq K(x, y, c) \\ 0 & \text{otherwise} \end{cases} \quad \text{and} \quad \sum_{k=1}^{\infty} \mathbf{M}^{(k)} = \mathbf{K}, \tag{3}$$

as shown in Figure 3 (left). With an arbitrary number of binary rollover masks, we can always render an updated modulo image by

$$\mathbf{I}_m^{(k)} = \mathbf{I}_m + 2^N \cdot (\mathbf{M}^{(1)} + \cdots + \mathbf{M}^{(k)}), \tag{4}$$

so we further transform Equation (2) into

$$P(\mathbf{K}|\mathbf{I}_m) = \prod_{k=0}^{\infty} P(\mathbf{M}^{(k+1)}|\mathbf{I}_m^{(k)}), \tag{5}$$

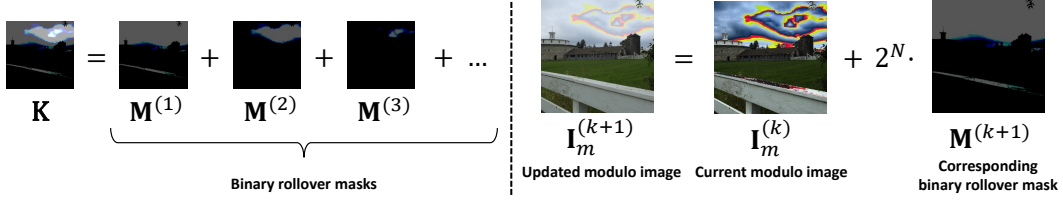

Figure 3: Left: the number of rollovers $\mathbf{K}$ can be decomposed into a series of binary rollover masks $(\mathbf{M}^{(1)}, \mathbf{M}^{(2)}, \mathbf{M}^{(3)}, \ldots)$. Right: a single iteration of the overall unwrapping pipeline in Equation (6). The number of rollovers $\mathbf{K}$ in the figure is linearly scaled to $[0, 255]$ for visualization, and $\mathbf{M}^{(k)}$ ($k = 1, 2, 3, \ldots$) uses the same scale factor as $\mathbf{K}$.

where $\mathbf{I}_m^{(0)} = \mathbf{I}_m$. To this end, estimating $P(\mathbf{M}^{(k+1)}|\mathbf{I}_m^{(k)})$ in $k$-th factor is equivalent to estimating the corresponding binary rollover mask given a modulo image, and the original problem becomes an iterative per-pixel binary labeling problem terminating when $\mathbf{M}^{(k+1)} = \mathbf{0}$.

As shown in Figure 2, UnModNet takes a single modulo image $\mathbf{I}_m$ as input, iteratively updates it by predicting the binary rollover mask $\mathbf{M}$, and outputs the HDR result $\mathbf{I}$ until the algorithm terminates. Each unwrapping iteration can be written as:

$$\mathbf{I}_m^{(k+1)} = \mathbf{I}_m^{(k)} + 2^N \cdot \mathbf{M}^{(k+1)} = \mathbf{I}_m^{(k)} + g(\mathbf{I}_m^{(k)}), \tag{6}$$

where $g$ represents the proposed UnModNet. An example is shown in Figure 3 (right).

## 3.2 Modulo edge separator

Despite the pixel intensity distribution of modulo regions is irregular, modulo images are always comprised of distinctive dense edges. Prominently, we recognize that a modulo camera brings abrupt intensity changes in continuous irradiance regions, resulting in modulo edges with large magnitude. Edges are effective cues for various image restoration tasks, such as reflection separation [29], moiré patterns removal [18], image inpainting [37], *etc.*, because the sparse nature of edges could relieve the ill-posedness of these problems. Similarly, we expect that a modulo edge separator could assist our goal of rollover mask prediction.

We first design a network module to predict channel-wise modulo edges $\mathbf{E}_m$ from a single modulo image $\mathbf{I}_m$, as shown in the first stage of Figure 2. Modulo edges $\mathbf{E}_m$, which encode boundary information about modulo regions, can be defined as $\mathbf{E}_m = \mathtt{bin}(\mathbf{E}_l - \mathbf{E}_n)$, where $\mathtt{bin}$ stands for binarization, $\mathbf{E}_l$ denotes the channel-wise edge map (edges of the modulo image $\mathbf{I}_m$), and $\mathbf{E}_n$ represents the intensity edges (edges of the ground truth HDR image $\mathbf{I}$). Since the modulo edges appear when "reset" of intensity from maximum to zero is triggered, its magnitude should be larger than most of the intensity edges. A simple verification is that by measuring the average edge magnitude of 3000 synthetic modulo images, we find that the magnitude of $\mathbf{E}_l - \mathbf{E}_n$ is around 4 times larger than $\mathbf{E}_n$.[2] This is helpful for the separation of $\mathbf{E}_m$ from $\mathbf{E}_n$. To better exploit this property, we propose to learn the residual between $\mathbf{E}_l$ and $\mathbf{E}_m$ instead of predicting $\mathbf{E}_m$ directly. Channel-wise Laplace kernels are used to filter the input modulo image $\mathbf{I}_m$ to obtain the edge map $\mathbf{E}_l$. Such a network can be described as:

$$\mathbf{E}_m = \mathbf{E}_l + g_e(\mathtt{cat}(\mathbf{E}_l, \mathbf{I}_m)), \tag{7}$$

where $g_e$ denotes the backbone network and $\mathtt{cat}$ stands for feature concatenation. In practice, we construct $g_e$ using an autoencoder [21] architecture with residual bottleneck blocks [19] to boost network depth, non-local operations [54] to enlarge receptive fields, and skip-connections to magnify the response of modulo edges.

Obtaining modulo edges $\mathbf{E}_m$ makes unwrapping much easier because modulo edges could be used as *a priori* which contains abundant boundary information about modulo regions. The modulo edges are jointly predicted for all channels, resulting in a more consistent estimation.

### 3.3 Rollover mask predictor

We have observed that modulo pixels are more likely to cluster in local regions, which is consistent with the fact that high dynamic range pixels are usually concentrated in small areas of an image (see Figure 1). This makes modulo regions distinctive from intensity regions since visually they show unnatural color appearances. Moreover, two modulo images with a difference of only one binary rollover mask (say $\mathbf{I}_m^{(k)}$ and $\mathbf{I}_m^{(k+1)}$) share similar structure patterns, *i.e.* $\mathbf{I}_m^{(k+1)}$ can be viewed as an updated modulo image whose maximum intensity is "one-period" (in our case 256) larger than $\mathbf{I}_m^{(k)}$.

We therefore design another network module to predict the binary rollover mask $\mathbf{M}$, given a modulo image $\mathbf{I}_m$ and its channel-wise modulo edges $\mathbf{E}_m$ as input, as shown in the second stage of Figure 2. Directly feeding the concatenation of $\mathbf{I}_m$ and $\mathbf{E}_m$ to the network makes the model hard to converge, because of the large domain gaps between the two types of data. To overcome this difficulty, we use convolutions and non-local blocks to extract the local and global features of $\mathbf{I}_m$ and $\mathbf{E}_m$, and fuse them with a concatenation and a squeeze-and-excitation (SE) block [23]. SE block learns normalized weights in each channel and recalibrates feature maps by re-weighting them. The predicted binary rollover mask produced from this module can be presented mathematically as follows:

$$\mathbf{M} = g_m(\texttt{SE}(\texttt{cat}(\mathcal{F}_i(\mathbf{I}_m), \mathcal{F}_e(\mathbf{E}_m)))), \tag{8}$$

where $g_m$ denotes the backbone network, $\texttt{SE}$ represents the SE block, $\mathcal{F}_i$ and $\mathcal{F}_e$ indicate the feature extraction processes for $\mathbf{I}_m$ and $\mathbf{E}_m$ respectively. As for $g_m$, we choose Attention U-Net architecture [40], and use residual bottleneck blocks and strided convolutions to substitute double convolution blocks and max-pooling layers in each scale respectively.

With the rollover masks becomes available, we can treat the unwrapping problem as an iterative per-pixel binary labeling problem as we have discussed in Section 3.1. With the semantic information provided by modulo images $\mathbf{I}_m$ and the boundary information provided by modulo edges $\mathbf{E}_m$, the estimation of the rollover mask tends to be more robust and the unwrapped image suffers less from modulus-intensity ambiguity.

### 3.4 Implementation details

**Loss function.** The total loss function of UnModNet is $\mathcal{L} = \alpha \cdot \mathcal{L}_e + \mathcal{L}_m$, where $\mathcal{L}_e$ defines the loss of the modulo edge separator, $\mathcal{L}_m$ defines the loss of the rollover mask predictor, and $\alpha$ is set to 1.0 empirically. The binary cross entropy loss is used for both $\mathcal{L}_e$ and $\mathcal{L}_m$.

**Dataset preparation.** Learning-based methods depend heavily on training data, but there is no existing dataset for our task. Therefore, we collect HDR images from a various of image and video sources [10, 11, 12, 13, 14, 27, 38, 57] and propose an effective dataset creation pipeline. The generation of the ground truth HDR image $\mathbf{I}$ can be expressed as $\mathbf{I} = \lfloor (2^B - 1) \cdot \texttt{clip}(\mathcal{E} \cdot \Delta t, [0, 1]) \rfloor$, where $B$ denotes the quantization bit depth, $\mathcal{E}$ indicates the relative irradiance values of each raw HDR image ($\mathcal{E} \in [0, 1]$), and $\Delta t$ is an appropriate exposure time to control the over-exposure rate.[3] The corresponding modulo image $\mathbf{I}_m$ and LDR image $\mathbf{I}_l$ can be calculated by Equation (1) ($N$ is set to 8 for 8-bit modulo images) and $\mathbf{I}_l = \texttt{clip}(\mathbf{I}, [0, 255])$ respectively. We choose $B = 12$ (*i.e.*, 12-bit HDR images with a maximum intensity 4095) and set the over-exposure rate between 5% and 30%. The images are resized and randomly cropped to $256 \times 256$ patches during the training process, and cropped to $512 \times 512$ patches for test.

**Training strategy.** We implement UnModNet[4] using PyTorch and apply a two-stage training strategy. First, to ensure a stable initialization of the training process, we train the modulo edge separator and rollover mask predictor independently for 400 and 200 epochs respectively. Then, we fix the modulo edge separator and train the entire network end-to-end for another 200 epochs. ADAM optimizer [26] is used with an initial learning rate $1 \times 10^{-4}$ for the first 200 epochs, and a linear decay to $5 \times 10^{-5}$ in the next 200 epochs. Dropout noise [48] and instance normalization [51] are added during training.

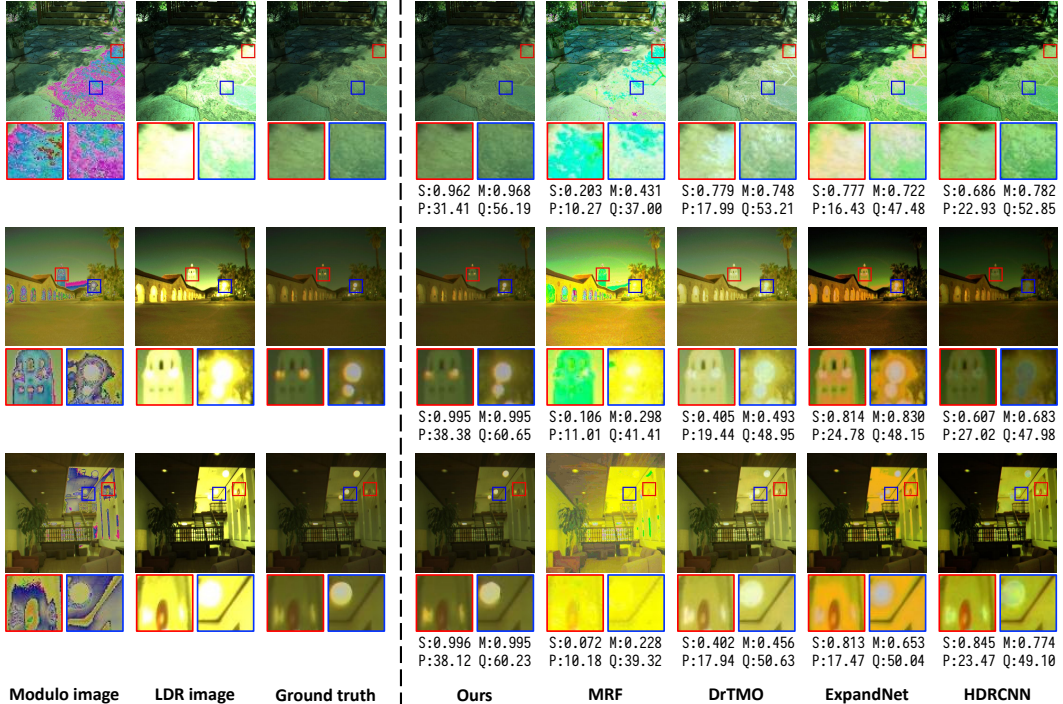

| Modulo image | LDR image | Ground truth | Ours | MRF | DrTMO | ExpandNet | HDRCNN |

Figure 4: Qualitative comparisons on synthetic data among UnModNet, the MRF-based unwrapping algorithm [59], and three state-of-the-art learning-based HDR reconstruction methods which take a single LDR image as input: DrTMO [7], ExpandNet [31], and HDRCNN [6]. Quantitative results evaluated using SSIM (S), MS-SSIM (M), PSNR (P), and Q-Score (Q) are displayed below each image.

Table 1: Quantitative evaluation results on synthetic data among UnModNet, the MRF-based unwrapping algorithm [59] (with fixed parameters), and three state-of-the-art learning-based HDR reconstruction methods which take a single LDR image as input: DrTMO [7], ExpandNet [31], and HDRCNN [6].

|         | Ours   | MRF [59] | DrTMO [7] | ExpandNet [31] | HDRCNN [6] |
|---------|--------|----------|-----------|----------------|------------|
| SSIM    | **0.937** | 0.177    | 0.663     | 0.724          | 0.721      |
| MS-SSIM | **0.925** | 0.332    | 0.685     | 0.690          | 0.759      |
| PSNR    | **28.51**  | 8.88     | 21.23     | 18.14          | 22.74      |
| Q-Score | **54.78**  | 36.35    | 51.87     | 48.04          | 47.70      |

## 4 Experiments

### 4.1 Evaluation on synthetic data

We compare the results of UnModNet to the MRF-based algorithm [59] which takes a single modulo image as input and three state-of-the-art learning-based HDR reconstruction methods which take a single LDR image as input: DrTMO [7], ExpandNet [31], and HDRCNN [6]. Since our method keeps the same set of parameters for all test cases, for a fair comparison, we fix the parameters of the MRF algorithm for evaluation as well. Note that comparing with learning-based single-image HDR reconstruction methods (DrTMO [7], ExpandNet [31], and HDRCNN [6]) might be a bit unfair because of the difference in types of input data (LDR image vs. modulo image), and we conduct such a comparison to show the effectiveness of using modulo images w.r.t. state-of-the-art single-image approaches. Visual quality comparisons of tone-mapped HDR images are shown in Figure 4[5]. Compared to the MRF-based algorithm using a modulo image, our model is robust under

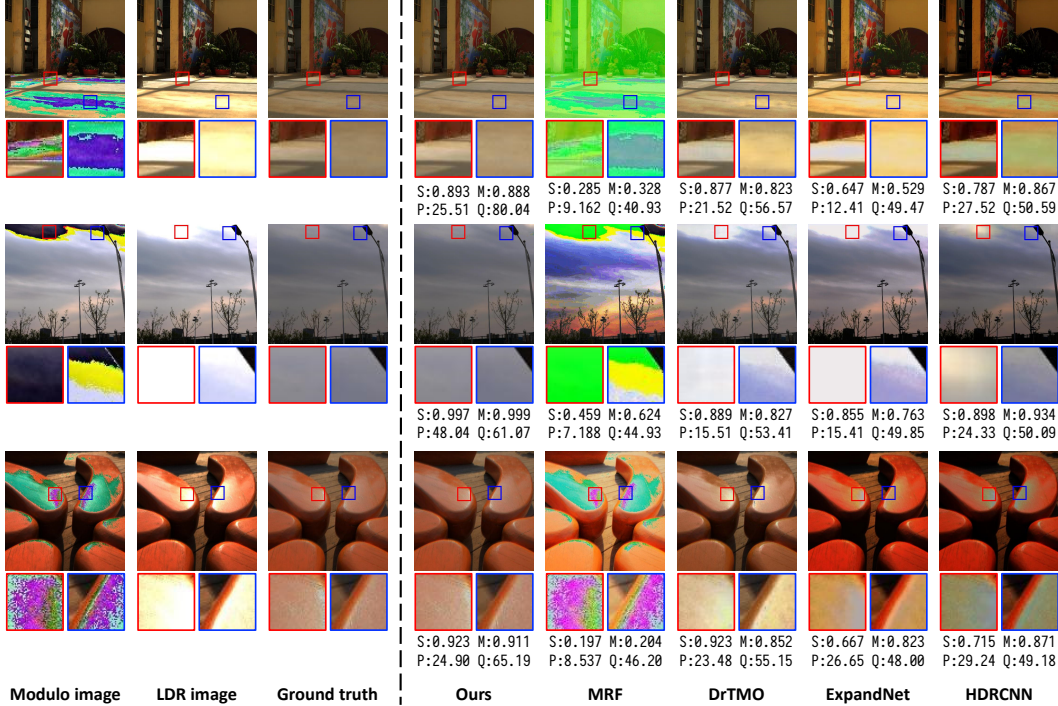

| | | | | | | | |
|---|---|---|---|---|---|---|---|
| | | | S:0.893 M:0.888 P:25.51 Q:80.04 | S:0.285 M:0.328 P:9.162 Q:40.93 | S:0.877 M:0.823 P:21.52 Q:56.57 | S:0.647 M:0.529 P:12.41 Q:49.47 | S:0.787 M:0.867 P:27.52 Q:50.59 |
| | | | S:0.997 M:0.999 P:48.04 Q:61.07 | S:0.459 M:0.624 P:7.188 Q:44.93 | S:0.889 M:0.827 P:15.51 Q:53.41 | S:0.855 M:0.763 P:15.41 Q:49.85 | S:0.898 M:0.934 P:24.33 Q:50.09 |
| | | | S:0.923 M:0.911 P:24.90 Q:65.19 | S:0.197 M:0.204 P:8.537 Q:46.20 | S:0.923 M:0.852 P:23.48 Q:55.15 | S:0.667 M:0.823 P:26.65 Q:48.00 | S:0.715 M:0.871 P:29.24 Q:49.18 |
| **Modulo image** | **LDR image** | **Ground truth** | **Ours** | **MRF** | **DrTMO** | **ExpandNet** | **HDRCNN** |

Figure 5: Qualitative comparisons on real RGB data. See the caption of Figure 4 for explanation.

strong local contrast or dense modulo fringes, while avoiding unwrapping incorrect regions and color misalignment. For example, the lighthouse (red box) in the middle row of Figure 4, which has drastic dynamic range changes, is correctly unwrapped by UnModNet, while the MRF-based algorithm fails to discriminate pixels in modulo from intensity regions and suffers from severe color misalignment artifacts. Compared to learning-based methods using an LDR image, our method performs better in recovering high contrast areas, and resembles the ground truth more closely. To evaluate the results quantitatively, we adopt four frequently-used image quality metrics including SSIM, MS-SSIM (multi-scale SSIM), PSNR, and Q-Score (produced by HDR-VDP-2.2 [35]). Results are shown in Table 1 (also for examples in Figure 4). Our model consistently outperforms the MRF-based and learning-based HDR reconstruction methods on all metrics. Furthermore, we evaluate the runtime of UnModNet on an NVIDIA 2080Ti GPU and the MRF-based algorithm on an Intel Core i7-8700K CPU (using a single core). Note that the MRF-based algorithm could not apply the GPU acceleration, so we can only run it on a CPU. At each iteration, UnModNet takes around $200ms$ to process a $512 \times 512$ modulo image, which is around 120 times faster than the MRF-based algorithm.

## 4.2 Evaluation on real data

**Modulo images from real RGB images.** We use a Fujifilm X-T20 mirrorless digital camera[6] to create a real dataset from RGB images. First, we take a series of images (around $7 \sim 9$) with bracketed exposures, and use the classical multi-image HDR reconstruction method [5] to merge them into an HDR image. The exposure value of each images are increased by 2 stops. Then, we use the dataset generation pipeline proposed in Section 3.4 to get the ground truth $\mathbf{I}$, the modulo image $\mathbf{I}_m$, and the corresponding LDR image $\mathbf{I}_l$. As shown in Figure 5[7], our model is able to reconstruct visually impressive HDR images with less artifacts and higher quantitative scores than other methods.

**Modulo images from a real sensor.** There are several technologies which could mod the scene radiance as a modulo image before converting to digital signals, such as digital-pixel focal plane array (DFPA) [3, 9, 50] (used in [59]), programmable readout circuit [33], and intelligent vision sensors (*e.g.*, Sony IMX500[8]), *etc*. We configure a retina-inspired fovea-like sampling model (FSM)

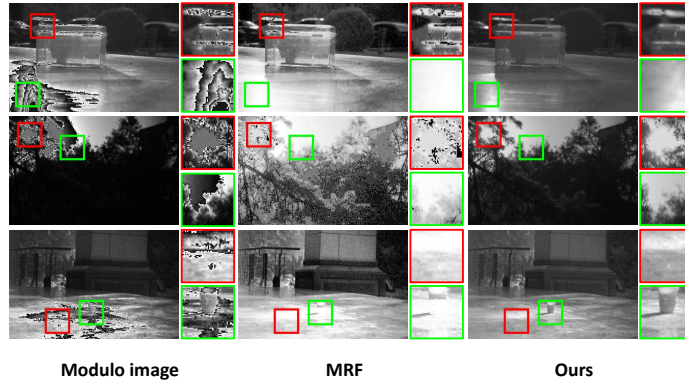

**Modulo image**  **MRF**  **Ours**

Figure 6: Qualitative comparisons on images captured by SpiCam-Mod between our model and the MRF-based algorithm [59].

Table 2: Quantitative evaluation results of ablation study.

|  | SSIM | MS-SSIM | PSNR | Q-Score |
|---|---|---|---|---|
| Directly predicting $\mathbf{K}$ | 0.739 | 0.743 | 21.82 | 47.50 |
| Removing modulo edge separator | 0.917 | 0.914 | 26.40 | 46.66 |
| Removing Laplace operation | **0.940** | 0.912 | 26.06 | 52.80 |
| End-to-end training | 0.931 | 0.924 | 27.08 | 49.62 |
| Our final model | 0.937 | **0.925** | **28.51** | **54.78** |

based spike camera [60] in modulo mode to act as a real modulo camera, named as *SpiCam-Mod*. SpiCam-Mod is able to measure luminance changes in each pixel as temporal spikes (0 and 1), and "reset" the number of spikes at each pixel whenever it reaches saturation.[9] Note that SpiCam-Mod has no color filter, so we can only provide results in grayscale as a proof of concept. As shown in Figure 6, our model outperforms the MRF-based algorithm [59] on recovering richer details of HDR contents.

### 4.3 Ablation study

To verify the validity of each model design choice, we conduct a series of ablation studies and show comparisons in Table 2. We first show the effectiveness of our iterative unwrapping pipeline by comparing with a model that directly predicts the number of rollovers $\mathbf{K}$. Then, we verify the necessity of the modulo edge separator by removing it and show the effectiveness of learning the residual between the edge map $\mathbf{E}_l$ and modulo edges $\mathbf{E}_m$ in the modulo edge separator by removing the Laplace operation. Finally, we validate the two-stage training strategy by training the entire network in an end-to-end manner.

## 5 Conclusion

We presented a learning-based framework for modulo image unwrapping to realize high dynamic range imaging. To deal with the ill-posedness of this problem, we reformulated it into a series of binary labeling problems and proposed UnModNet to iteratively estimate the binary rollover masks of an input modulo image. Our model design solved some fundamental issues in the previous MRF-based algorithm [59], including modulus-intensity ambiguity, strong local contrast, and color misalignment.

The highest bit depth that can be achieved by the existing model is constrained by the configuration of training data.[10] As future work, we plan to extend UnModNet to support dynamic bit depth.

## Broader Impact

Our research is about a new camera framework that aims to capture high-quality HDR images. It could be integrated into the image processing pipeline of camera sensors to improve the ability of recording scenes with a very high dynamic range. The users of mobile cameras may benefit from this research because they could conveniently take photos without being annoyed by over- or under-exposure artifacts. Besides, it might be helpful to build a scientific imaging system that needs to record high dynamic range scenes, such as astronomy and microscope cameras.

Although the modulo camera-based framework could theoretically achieve unbounded dynamic range, its generalization capability is limited by the diversity of the training data. The unwrapping algorithm may fail when the captured scene has a very high dynamic range which exceeds the maximum dynamic range of the images in the training data by a large margin. If that happens in a large region of pixels, we would recommend using LDR images instead since they have more natural color appearances.

## Acknowledgments and Disclosure of Funding

This work was supported in part by National Natural Science Foundation of China under Grant No. 61872012, No. 61876007, National Key R&D Program of China (2019YFF0302902), Beijing Academy of Artificial Intelligence (BAAI), Beijing major science and technology projects (Z191100010618003), and Australian Research Council Grant DE-180101438.

## Footnotes

[2]Please refer to the supplementary material for more details about modulo edges and experimental validation.

[3]More details about the dataset creation pipeline can be found in the supplementary material.

[4]Detailed network architecture can be found in the supplementary material.

[5]More synthetic results can be found in the supplementary material.

[6]https://fujifilm-x.com/global/products/cameras/x-t20/

[7]More real RGB results can be found in the supplementary material.

[8]https://www.sony.net/SonyInfo/News/Press/202005/20-037E/

[9]Please refer to the supplementary material for how we use SpiCam-Mod to capture modulo images.

[10]We demonstrate 16-bit HDR reconstruction results in the supplementary material.

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
