[Supplementary Material]

# Supplementary Material
# UnModNet: Learning to Unwrap a Modulo Image for High Dynamic Range Imaging

**Chu Zhou**[1] **Hang Zhao**[2] **Jin Han**[1] **Chang Xu**[3] **Chao Xu**[1] **Tiejun Huang**[4,5] **Boxin Shi**[4,5*]

[1]Key Lab of Machine Perception (MOE), Dept. of Machine Intelligence, Peking University
[2]Massachusetts Institute of Technology
[3]School of Computer Science, Faculty of Engineering, University of Sydney
[4]NELVT, Dept. of Computer Science, Peking University
[5]Institute for Artificial Intelligence, Peking University
{zhou_chu, hanjin1619, tjhuang, shiboxin}@pku.edu.cn,
zhaohang0124@gmail.com, c.xu@sydney.edu.au, xuchao@cis.pku.edu.cn

## 6 More details about modulo edges

In this section, we provide more details about the modulo edges, including the visualization of the definition of modulo edges and the experimental validation of "the magnitude of modulo edges should be larger than most of the intensity edges", corresponding to Footnote 2 in Section 3.2 of the paper.

As shown in Figure 7 (left), modulo edges $\mathbf{E}_m$ can be defined as:

$$\mathbf{E}_m = \texttt{bin}(\mathbf{E}_l - \mathbf{E}_n),\tag{1}$$

where `bin` stands for binarization, $\mathbf{E}_l$ denotes the channel-wise edge map (edges of the modulo image $\mathbf{I}_m$), and $\mathbf{E}_n$ represents the intensity edges (edges of the ground truth HDR image $\mathbf{I}$). We can see that the edge map $\mathbf{E}_l$ of a modulo image can be labeled as modulo edges $\mathbf{E}_m$, which encode boundary information about modulo regions, and intensity edges $\mathbf{E}_n$, which are redundant in our task.

As shown in Figure 7 (right), by measuring the average edge magnitude of 3000 synthetic modulo images, we find that the magnitude of $\mathbf{E}_l - \mathbf{E}_n$ is around 4 times larger than $\mathbf{E}_n$, which provides a strong prior to predict the modulo edges $\mathbf{E}_m$.

Figure 7: Left: the relationship between the edge map $\mathbf{E}_l$, modulo edges $\mathbf{E}_m$, and intensity edges $\mathbf{E}_n$. Right: the statistics of the average edge magnitude of $\mathbf{E}_l - \mathbf{E}_n$ and $\mathbf{E}_n$ (scaled to $[0, 255]$).

---

# 7 Dataset creation pipeline in detail

In this section, we explain the proposed dataset creation pipeline in detail, corresponding to Footnote 3 in Section 3.4 of the paper.

For each raw HDR image, we first resize and randomly crop it to an $X \times X$ ($X$ is set to $512$ during the training process and set to $256$ for test) image patch $\mathbf{I}_r$ for data augmentation, and scale it to relative irradiance values $\mathcal{E} \in [0, 1]$:

$$\mathcal{E} = \texttt{scale}(\mathbf{I}_r, [0, 1]). \tag{2}$$

Next, we re-expose $\mathcal{E}$ to an appropriate exposure time $\Delta t$ by controlling the over-exposure rate of the image in the range of $[5\%, 30\%]$, which assumes that people usually do not take or save photos that contain largely over-exposed areas:

$$\mathcal{E}' = \texttt{clip}(\mathcal{E} \cdot \Delta t, [0, 1]). \tag{3}$$

Then, we quantize $\mathcal{E}'$ to $B$ bits as the ground truth HDR image $\mathbf{I}$:

$$\mathbf{I} = \left\lfloor (2^B - 1) \cdot \mathcal{E}' \right\rfloor. \tag{4}$$

As the ground truth HDR image $\mathbf{I}$ becomes available, we could compute other variables easily:

(1) The corresponding 8-bit modulo image $\mathbf{I}_m$ and the number of rollovers per pixel can be obtained by:

$$\mathbf{I}_m = \texttt{mod}(\mathbf{I}, 256) \quad \text{and} \quad \mathbf{K} = \frac{(\mathbf{I} - \mathbf{I}_m)}{256}, \tag{5}$$

where $\mathbf{K} = \{K(x, y, c)\}$ denotes the number of rollovers per pixel, $(x, y)$ is the pixel coordinate, and $c$ is the color channel index.

(2) The binary rollover masks can be acquired by:

$$\sum_{k=1}^{\infty} \mathbf{M}^{(k)} = \mathbf{K} \quad \text{and} \quad M^{(k)}(x, y, c) = \begin{cases} 1 & \text{if } k \leq K(x, y, c) \\ 0 & \text{otherwise} \end{cases}, \tag{6}$$

where $\mathbf{M}^{(k)} = \{M^{(k)}(x, y, c)\}$ represents the $k$-th ($k = 1, 2, 3, \ldots$) binary rollover mask.

(3) The updated modulo images can be attained by:

$$\mathbf{I}_m^{(k)} = \mathbf{I}_m + 2^N \cdot (\mathbf{M}^{(1)} + \cdots + \mathbf{M}^{(k)}), \tag{7}$$

where $\mathbf{I}_m^{(k)}$ stands for the $k$-th ($k = 1, 2, 3, \ldots$) updated modulo image.

(4) The LDR image $\mathbf{I}_l$ for test can be calculated by:

$$\mathbf{I}_l = \texttt{clip}(\mathbf{I}, [0, 255]). \tag{8}$$

To this end, our goal is trying to find an appropriate exposure time $\Delta t$ that controls the over-exposure rate in a proper range. The over-exposure rate $r_o$ can be defined as the ratio of the number of over-exposed pixels to the total number of pixels. According to Eq. (3) and Eq. (4), we could compute $r_o$ using relative irradiance values $\mathcal{E}$:

$$r_o = \frac{\texttt{numel}(\mathcal{E}_o)}{\texttt{numel}(\mathcal{E})}, \tag{9}$$

where $\texttt{numel}$ stands for the operation of counting the number of array elements, and $\mathcal{E}_o$ denotes the corresponding relative irradiance values of the over-exposed pixels. $\mathcal{E}_o$ should satisfy:

$$\mathcal{E}_o \geq \frac{256}{\Delta t \cdot (2^B - 1)}. \tag{10}$$

We could apply a binary search to achieve this, as shown in Algorithm 1 below. The upper limit $m$ of the number of iterations is set to $15$ in our case.

**Algorithm 1:** The algorithm of finding an appropriate exposure time $\Delta t$

**Input:**

     The relative irradiance values $\mathcal{E} \in [0, 1]$;

     The quantization bit depth $B$;

     The given range $[a, b]$ of the over-exposure rate;

     The upper limit $m$ of the number of iterations.

**Output:**

     An appropriate exposure time $\Delta t$.

```
   /* Initialization                                                      */
 1 n ← 0;                                    // Initialize the number of iterations
 2 l ← 0;                        // Initialize the lower bound of the search space
 3 u ← 1;                        // Initialize the upper bound of the search space
   /* Start the binary search                                             */
 4 while c < m do
 5     if n = 0 then
 6         v ← rand([l, u]);   // Randomly initialize a value v to denote  256/(Δt·(2^B−1))
 7     end
 8     else
 9         v ← (l+u)/2;                                       // Renew the value v
10     end
11     E_o ← filter_with_condition(E ≥ v);                   // Filter E to get E_o
12     r_o ← numel(E_o)/numel(E);                     // Calculate the over-exposure rate r_o
13     if r_o > b then
14         l ← v;          // r_o is too large, try to decrease it by increasing v
15     end
16     else if r_o < a then
17         u ← v;          // r_o is too small, try to increase it by decreasing v
18     end
19     else
20         Δt ← 256/(v·(2^B−1));        // r_o is in the given range [a, b], calculate Δt
21         break;
22     end
23     n ← n + 1;
24 end
   /* Decide whether to discard or retain                                 */
25 if n = m then
26     return; // Discard since we cannot find a satisfied Δt in m iterations
27 end
28 else
29     return Δt;        // Return the satisfied exposure time Δt we have found
30 end
```

# 8 Network architecture in detail

In this section, we present the architecture details of our proposed UnModNet, as shown in Figure 8, Figure 9 and Figure 10, corresponding to Footnote 4 in Section 3.4 of the paper.

Figure 8: Backbone network of modulo edge separator ($g_e$) in detail.

Figure 9: Backbone network of rollover mask predictor ($g_m$) in detail.

Figure 10: Feature extraction processes for modulo image $\mathbf{I}_m$ and modulo edges $\mathbf{E}_m$ ($\mathcal{F}_i$ and $\mathcal{F}_e$ respectively) in detail.

# 9 More synthetic results

In this section, we provide more qualitative comparisons on synthetic data among UnModNet, the MRF-based unwrapping algorithm [4] and three state-of-the-art learning-based HDR reconstruction methods which take a single LDR image as input: DrTMO [2], ExpandNet [3], and HDRCNN [1], as shown in Figure 11, Figure 12, Figure 13, Figure 14, and Figure 15, corresponding to Footnote 5 in Section 4.1 of the paper.

Figure 11: Qualitative comparisons on synthetic data: part 1.

Figure 12: Qualitative comparisons on synthetic data: part 2.

Figure 13: Qualitative comparisons on synthetic data: part 3.

Figure 14: Qualitative comparisons on synthetic data: part 4.

Figure 15: Qualitative comparisons on synthetic data: part 5.

# 10 More real RGB results

In this section, we provide more qualitative comparisons on real RGB images among UnModNet, the MRF-based unwrapping algorithm [4] and three state-of-the-art learning-based HDR reconstruction methods which take a single LDR image as input: DrTMO [2], ExpandNet [3], and HDRCNN [1], as shown in Figure 16, corresponding to Footnote 7 in Section 4.2 of the paper.

Figure 16: Qualitative comparisons on synthetic data.

## 11 SpiCam-Mod workflow

In this section, we demonstrate how we use the retina-inspired fovea-like sampling model (FSM) based spike camera [5] operating in modulo mode, a.k.a. SpiCam-Mod, to capture modulo images, corresponding to Footnote 9 in Section 4.2 of the paper.

Unlike conventional frame-based cameras that accumulate all the photoelectric information within an exposure window to form a snapshot image, each pixel of an FSM-based spike camera responds to the incoming light independently and fires a stream of spikes to record the time-varying irradiance changes. The formation of a spike can be expressed as an "accumulate-fire-reset" cycle: The accumulator of each pixel accumulates the electric charges $\mathcal{I}(t)$ of the irradiance digitalized by an A/D converter:

$$A(t) = \int_0^t \mathcal{I}(x)dx, \tag{11}$$

where $A(t)$ denotes the accumulated electric charges. Once $A(t)$ reaches a dispatch threshold $\tau$, a spike is fired at this time stamp. This signal also resets the corresponding accumulator, in which all the electric charges are drained (*i.e.*, resets $A(t)$ to zero). Pixels on the image sensor may fire spikes in arbitrary time, while the camera reads out the spike streams as a discrete-time binary signal. Specifically, the sensor checks the accumulators periodically within a fixed interval. If a spike is fired, it reads out signal 1 and resets the accumulator of this pixel immediately, otherwise it reads out signal 0 at this time stamp, and we obtain a spike image $\mathbf{S} = S(x, y, c)$ ($x \in [1 .. 400], y \in [1 .. 250], c = 1$ for currently available spike camera) by counting the spikes pixel-wisely. Please refer to [5] for more details.

We could further program the FSM-based spike camera firmware to configure it in modulo mode which captures modulo images as follows:

(1) Set a threshold $V$ for each spike image;

(2) Let $S(x, y, c)$ equal to zero once it reaches $V$ and re-count spikes periodically;

(3) Scale $\mathbf{S}$ to relative irradiance $\mathcal{E} \in [0, 1]$ and quantize it to an 8-bit modulo image $\mathbf{I}_m$.

The threshold $V$ varies for scenes with different dynamic ranges. Since an appropriately exposed modulo image does not contain large modulo regions, we set $V$ to make sure that the proportion of modulo pixels is in the range of $[5\%, 30\%]$, which is similar to the pipeline proposed in Section 7.

## 12  16-bit HDR reconstruction results

In this section, we provide qualitative comparisons on 16-bit data among UnModNet, the MRF-based unwrapping algorithm [4] and three state-of-the-art learning-based HDR reconstruction methods which take a single LDR image as input: DrTMO [2], ExpandNet [3], and HDRCNN [1], as shown in Figure 17, corresponding to Footnote 10 in Section 5 of the paper.

As mentioned in Section 5 of the paper, to deal with HDR reconstruction with higher precision, we only need to create another training dataset using HDR images stored with higher precision by rendering ground truth HDR images with a higher bit depth (changing $B$ in Equation (3) from 12 to 16 while keeping other settings the same). We can see that the proposed UnModNet could handle 16-bit HDR reconstruction robustly and generate visually impressive HDR images. However, sometimes the details in regions with a very high dynamic range are not as rich as the previous 12-bit results, because of the inherent difficulty in recovering modulo images with a much larger number of binary rollover masks (the modulo image of a 16-bit HDR image has 255 binary rollover masks, while the 12-bit counterpart has only 15).

Figure 17: Qualitative comparisons on 16-bit data.