[Reviews · NeurIPS 2020]

Review 1

Summary and Contributions: This paper presents a deep learning solution to the problem of processing an image from a "modulo camera". This is a highly experimental camera in which the sensor "wraps around" instead of saturating, which theoretically gives it an unbounded dynamic range. This paper shows how to frame this problem such that it can be solved with neural networks.

Strengths: As someone who is interested in computational imaging, I find modulo cameras to be quite interesting and compelling. The proposed method seems reasonable to me. The "rollover" structure and the heuristics that are used to inform the treatment of edges and "fringes" seem solid and insightful to me. The paper very clearly demonstrates that the proposed model beats the baseline from [55] (which appears to be based on graph cuts) by a very large margin.

Weaknesses: My primary concern with this paper is that the problem it is addressing is *extremely* niche --- Modulo cameras are a somewhat obscure problem even within the realm of the computational imaging community. If I was reviewing this paper for a computational imaging/photography conference, I would be more charitable towards this paper. But this subject is unlikely to be of interest to the general NeurIPS audience, and this paper seems unlikely to reach its intended audience if presented at NeurIPS. And the specifics of this neural network architecture are so specifically tailored to this particular problem that I'm not sure what a general ML researcher could come away from this paper with, nor am I convinced that this is a problem that should be popularized with ML researchers as, again, a solution to this problem has limited practical value given that modulo cameras are still a largely hypothetical concept. My other concern with this paper (which would be a significant concern even if I were reviewing this paper in a computational imaging conference) is that the baseline evaluation is misleading. The most important comparison is to [55], which is the paper that originates the idea of a modulo camera, and which the evaluation suggests is outperformed by an enormous margin (I'm actually a bit confused as to why this margin is so significant as the images shown in [55] look much better than the images shown here in the evaluation against [55], but I'm willing to believe that the images in [55] were cherry-picked). This evaluation would have been easier to parse if the authors had simply used some of the images presented in [55], so at least a qualitative comparison could have been made. But my biggest concern is comparison with the "baseline" techniques in Table 1 other than [55], which are *not for the modulo camera task*! The techniques of [7, 30, 6] are learning-based methods for regressing from an LDR image *from a conventional camera* to an HDR image --- they are not techniques for processing modulo images. The input to the baseline techniques is not the same input as is used by the proposed model. This means that this baseline evaluation is not actually evaluating the accuracy of the proposed model, it is evaluating the accuracy of a modulo camera versus a conventional camera. I do not see the point of this evaluation outside of a computational imaging context. The actual evaluation that I would like to see is against other basic neural networks, or against other non-learned techniques for processing modulo imagery. I am not sure very many such techniques exist because (as previously stated) this problem is not well-studied, but a quick search let me find three papers that appear to propose solutions to the problem of [55]: “Robust Multi-Image HDR Reconstruction for the Modulo Camera”, “Reconstruction from Periodic Nonlinearities, with Applications to HDR Imaging”, and “Signal Reconstruction from Modulo Observations”. Or, the baseline evaluation could be performed by applying conventional CNNs (U-Nets, etc) to the modulo imaging tasks, as this would show that the architecture presented here is indeed necessary for this task. But I see very little value in having "baseline" comparisons in which the input comes from a completely different kind of camera as the images used as input to the proposed model, if the goal of the evaluation is to show the value of the model (and not the kind of camera).

Correctness: I didn't see any issues here, except for my previously describe concerns regarding the baseline evaluation using conventional LDR images instead of modulo images.

Clarity: The writing was a little rough at times but generally acceptable. Some typos I noted: “The real-world scene has” -> “Real-world scenes have” “Post-processed with a customized Markov random field (MRF) based algorithm, it can practically restore the HDR image by unwrapping the captured modulo image” -- Nongrammatical Typo in the bibtex: “Internatoinal” “channel-wisely;” -> “channel-wise;”?

Relation to Prior Work: Yes

Reproducibility: Yes

Additional Feedback: After reading the rebuttal and discussing with the other reviewers, my feelings towards this paper have not changed significantly.


Review 2

Summary and Contributions: The paper proposes a deep-leaning based technique for 2D phase unwarping of modulo HDR images. The authors demonstrate an improvement over an MRF-based solution.

Strengths: (S1) Although the idea of modulo HDR camera is not novel, the formulation of the problem as a binary labeling problem is novel and interesting. The ablation study justifies the approach. (S2) The proposed technique seems to outperform the previous work (but see also W1 below) (S3) The paper is well written and presented, the evaluation is solid, compared with a good number of methods (but see also W2) (S4) Tested with both 12 and 16-bit modulo 256 cameras. (S5) Tested with the "spike" camera.

Weaknesses: (W1) The results for the MRF-based method [55] look suspiciously bad. The results shown in the original paper look much better for similar images. Yet, the current submission does not include any of the standard HDR images that are also shown in [55]. Are those results generated with the original implementation? (W2) Could any of the 2D phase unwarping methods based on deep-learning, mentioned in Related Work, be used to reconstruct an HDR image? If not, it should be explained. If yes, the comparison with one of such methods should be included. (W3) The method does not consider the physical limitations of such modulo cameras: noise characteristic and blooming on the senor. A proper camera noise model, for example one from: Aguerrebere, C., Delon, J., Gousseau, Y., & Musé, P. (2013). Study of the digital camera acquisition process and statistical modeling of the sensor raw data. could be used to generate synthetic data. (W4) The claim that the method is 120 times faster than modulo unwarping is not entirely fair: does the compared algorithm run on the same GPU? The details on the MRF implementation should be included. (W5) The choice of 8-bits for modulo image seems arbitrary - most sensors are equipped with at least a 12-bit ADC. But the actual motivation for the modulo camera should be the limited full well capacity. It would be good if the choice of the "modulo" was linked to the physical properties of the sensor. (W6) The SSIM and PSNR results should not be computed on the linear irradiance values. Those should be used with logarithmic or PU-transforms (see for example https://doi.org/10.1117/12.765095) (W7) Can the method process images larger than 512x512?

Correctness: I am overall positive about this paper, as it offers an interesting solution and it shows good quality of the work. Weaknesses (W4-W7) are minor and can be addressed in a revision. W3 is future work. However, I would ask the authors to address (W1-W2) in their response.

Clarity: The paper is well written.

Relation to Prior Work: The authors should better explain the relation to 2D phase unwarping (W2)

Reproducibility: Yes

Additional Feedback: * Please explain how the images were tone mapped to show them in the paper. Thank you for clarifying the reason for poor performance of [55] in the authors' feedback.


Review 3

Summary and Contributions: This paper proposes a two-stage network to recover a high dynamic range (HDR) image from its modulo measurements. The two-stage supervised training, with intermediate supervision on modulo edges, is intended to make the task easier for the network to learn. This paper demonstrates better performance than the comparing methods.

Strengths: + The problem formulation which decomposes the rollover mask K into several binary roller masks M appears to be correct. + This paper provides a good set of comparisons with other methods on different images using different metrics. + This paper falls under the area of computational photography, which is one of the NeurIPS subject area. Modulo sensors have been proposed in recent years, but these sensors are not as widely used as conventional high dynamic range sensors.

Weaknesses: - The pros and cons of different methods are not comprehensively analyzed. In section 2, Related Work - Single-image HDR reconstruction, this paper just listed some state-of-the-art methods without specifically stating the advantage and disadvantage of them. - This paper claims that the proposed method runs 120 times faster than the previous MRF-based algorithm. However, it does not report the run-time of the comparing methods. - Minor issues: -- Eqn. 6 and Fig. 2 do not match. -- The dashed-line box of “UnModNet” should not include the \plusdot on the right hand side of the “Rollover mask predictor”.

Correctness: The claims, method, and experiments appear to be correct.

Clarity: This paper is well written. The graphs and equations are clear and easy to understand.

Relation to Prior Work: This paper mainly compares with the MRF-based unwrapping algorithm [55]. These comparisons are empirical but not analytical. This paper does not provide the detailed analysis on the differences between the previous works.

Reproducibility: Yes

Additional Feedback: After reading the author's feedback and comments from other reviewers, it seems to me that the parameters for [55] were not properly tuned for different images, which makes the comparison unfair. The method proposed in this paper is for modulo cameras, but Table 1 and Figure 4 compare with methods that recover high dynamic range (HDR) images from low dynamic range (LDR) images. A modulo camera will not provide an LDR image, so this comparison is not very informative.


Review 4

Summary and Contributions: This paper proposes a learning based approach for Modulo image unwrapping. Modulo image only restores the modulo rather than the absolute values of image luminance, and therefore, it can store 12bit illuminance in a 8bit storage. Most importantly, physically, it allows capturing exposure in a much larger dynamic range without senor under-flow or over flow. Modulus images need to be unwrapped/reconstructed for further utilization and visualization. Existing methods are rule based. And this work proposes a deep neural network based approach. Three customized model designs (modulo edge separator, rollover mask predictor, and consistent color prediction) were proposed to solve the three fundamental issues (modulus-intensity ambiguity, strong local contrast, and color misalignment) in the previous MRF-based unwrapping algorithm [55]. A real modulo camera based on a spike camera was employed to prove the proposed concept. Experimental results show that the overall performance is superior to the previous MRF-based unwrapping algorithm and several state-of-the-art learning-based HDR reconstruction methods which take a single LDR image as input.

Strengths: The idea to use deep network for phase unwrapping is novel and interesting. The idea of using a modulo camera for HDR was proposed 5 years ago, which had unique advantages in realizing “unbounded” HDR using a single image. However, the original solution of resolving an MRF problem without data term and hand-crafted priors was quite fragile. This paper is the first work that learns the unwrapping process of a modulus image. By reformulating the problem as a series of binary labeling problems and iteratively estimating the binary rollover mask of the input modulo image, which is a novelty point of view, the problem can be solved in a much more stable manner. All modules of UnModNet (modulo edge separator, rollover mask predictor, and consistent color prediction) are carefully and specially designed to deal with the problem of unwrapping a modulus image, which means the authors do spend efforts in observing and analyzing the properties of modulus images. The idea is clearly presented and the experiments are sufficient and reasonable. The performance improvement is significant according to its own reporting. Sufficient experiment results validate that the proposed model designs successfully solved the three fundamental issues (modulus-intensity ambiguity, strong local contrast, and color misalignment) in the previous MRF-based algorithm. In addition to extensive synthetic and real simulation (using multi-bracketed methods to capture real data) experiments, this paper adopted a retina-inspired fovea-like sampling model (FSM) based spike camera to reconfigure it as a modulo camera, which makes the verification convincing and suggests an alternative solution for real modulo sensor.

Weaknesses: The paper mentions there is no existing dataset for the task and therefore has created its own dataset. However, I do not find much information about this dataset. The performance improvement is quite significant, however, it is better to show some evidence that it is not overfitting. Is there any evidence of general applicability? Since this paper targets at a quite unique problem, intuitive illustrations are important for people who are unfamiliar with this topic to follow easily. For example, the 3D color bar in Figure 1 is not easy to understand, it would be better to have some extra explanation or just use a 2D one instead.

Correctness: Theoretical proof of the concept is provided and is reasonable.

Clarity: This paper clearly analyzed the fundamental issues of the previous MRF-based algorithm and proposed a new unwrapping pipeline and three customized model designs to deal with them.

Relation to Prior Work: Yes.

Reproducibility: Yes

Additional Feedback: The authors should release the code upon acceptance for reproducibility.

[Author Response · NeurIPS 2020]

We sincerely thank all reviewers for their valuable comments and suggestions. We respectfully think it should be
interesting to share with the NeurIPS community how to learn unwrapping a natural image, though the camera might be
obscure. We will fix the typos and improve the figures as suggested in the final version. Below we respond to specific
comments and concerns.

**R1&R2: Results of the MRF algorithm [55].** Those results are
indeed generated with the original implementation (we have got the
code from the authors). We noticed and confirmed with the authors that
the parameters of the MRF algorithm (especially the energy function
and the cliques to consider) need carefully fine-tuning on different
input data to achieve optimal results, and the fine-tuning process is quite tricky. Moreover, for RGB data, it independently
unwraps each channel with different parameters. According to our experiments, the images shown in [55] should be
"cherry-picked", as mentioned by R1. Since our method uses the same set of parameters for all test cases, for a fair
comparison, we fix the parameters of the MRF algorithm for evaluation as well. As shown in the right figure (examples
are from Figure 4 and 9 of [55]), by applying the MRF algorithm using the same set of parameters, the left image is
well recovered while the right one is not. In comparison, our learning-based method gives consistently good results.
**R1: Other non-learned techniques for processing modulo imagery.** Please kindly note that our method only takes a
*single* modulo image as input. We did not find other suitable baselines. Papers mentioned in the review "Robust Multi-
Image HDR Reconstruction for the Modulo Camera", "Reconstruction from Periodic Nonlinearities, with Applications
to HDR Imaging", "Signal Reconstruction from Modulo Observations", all require multiple images as input (what's
more, the model presented in the first paper is limited to two modulo periods). Multiple modulo images could be
merged for HDR as done in [55], but such a multi-image approach is beyond the scope of our paper. We will clarify this
assumption in the final version, and consider learning to unwrap multiple modulo images as our future work.
**R1: Evaluations on other conventional CNN models.** We respectfully point out that we have included such results in
our ablation study. For example, we have tested a vanilla attention U-Net structure that predicts the number of rollovers
$K$ (referring to "Directly predicting $K$" in Table 2 of the paper). To show the effectiveness of modulo edge separator, we
have also tried a single rollover mask predictor to directly output binary rollover mask (referring to "Removing modulo
edge separator" in Table 2 of the paper). The final rollover mask predictor adopts a modified attention U-Net structure,
and it outperforms a vanilla attention U-Net structure by 2.21 dB for PSNR according to our previous experiments
(when we were designing our network). We will include the results of such a vanilla attention U-Net in the final version.
**R2: Other deep-learning based 2D phase unwrapping methods.** We have tried our best to search for the code of
such methods, *e.g.*, [41, 44, 49], but none of them is available now. We are afraid that re-implementing them within
the rebuttal period could not produce reliable results. We will try to include such a comparison in the final version.
However, we suspect these methods designed for phase maps instead of modulo images could not compete with either
[55] or our method, due to the essentially different natures of phase maps and wrapped natural images.
**R2: Details about the tone mapping method.** We use the OpenCV-Python's Reinhard tone mapping algorithm
(`cv2.createTonemapReinhard(intensity=-1.0, light_adapt=0.8, color_adapt=0.0)`) for visualization.
**R2&R3: Implementation details of the MRF algorithm [55].** The code of [55] is written in MATLAB script and
cannot run on GPU, so we run it on an Intel Core i7-8700K CPU (using a single core), while UnModNet runs on an
NVIDIA 2080 Ti GPU (thanks to the PyTorch's CUDA acceleration). At each iteration, UnModNet takes around 200ms
to process a $512 \times 512$ modulo image, and [55] takes around 24s.

**R2: Evaluation metrics (SSIM, PSNR, MS-SSIM).** We have further conducted
evaluations on the tone-mapped HDR images with metrics SSIM, PSNR, and MS-
SSIM, and the advantage of our method did not change. For example, the PSNR
(dB) results in Table 1 of the paper become 30.33 for our method, 7.73 for MRF
[55], 19.99 for DrTMO [7], 18.79 for ExpandNet [30], and 18.68 for HDRCNN [6].
We will include complete results in our final version.
**R2: High-resolution results.** Examples (scaled to the same height) are shown in the right figure.
**R2: Physical limitations and properties of the sensor.** The robustness might be improved if we include a proper
camera noise model in our dataset generation pipeline, and we consider to add it in our future work. Our method could
adapt to other bit depth (*e.g.*, 12-bit) if we change the bit depth of the modulo images in the training dataset.
**R3: Pros and cons of different single-image HDR reconstruction methods.** We will improve the the related work
section by comprehensively discussing pros and cons of each method.
**R4: Details of our dataset.** Our training dataset consists of 21312 pairs of modulo and updated (referring to Equation
(1) of the paper) image patches in $256 \times 256$ resolution, which are randomly cropped and generated from $600$ HDR
images collected from various sources [12, 11, 13, 14, 37, 53, 10, 27]. The details of our dataset creation pipeline and
the parameter settings could be found in Section 7 of the supplementary material.
**R4: Evidence of not overfitting and the generalization applicability.** The evaluation on two types of real captured
data could prove that our method does not overfit, and has a good generalization ability.

[Meta-Review · NeurIPS 2020]

The submission has received two positive and two negative reviews. The post-rebuttal discussion has not lead to convergence, and the opinion of the reviewers remain split. The concerns of the "negative" reviewers are: 1) The application is too niche (R1). However, the topic of the paper falls into NeurIPS call for papers, as it is related to low-level computer vision, compressed sensing, deep neural architectures. 2) The comparison to [55] may be invalid since the qualitative performance of [55] in the submission seems to be considerably worse than in the original paper. The authors rebut that the results in [55] were cherry-picked and that they use the code from [55], while fixing the parameters. There is a quantitative comparison with [55] suggesting that the new method is better. In the absence of evidence to the contrary, the rebuttal seems to be plausible. 3) There are other prior works, to which comparisons should have been implemented. The authors rebut that those methods take multiple images and are hence not comparable to the proposed one. 4) The comparison in Table 1 is misleading. This is indeed a concern, but does not constitute grounds for rejection in the opinion of the area chair. 5) The runtime comparison is misleading since CPU and GPU runtimes are compared. Again, this is a valid criticism, but it can be clarified in the final version without major revision of the paper and its claims. Overall, the negative reviews do not seem to provide grounds for rejection. Given the support of the two positive reviewers, the suggestion of the area chair is to accept. In the final version, the authors should incorporate the feedback of the reviewers. In particular, the comparison in Table 1 and Figure 4 should spell very clearly that the compared methods are not truly comparable and that the comparison is performed due to the lack of more similar prior art. Table 1 caption must be expanded. Furthermore, as promised in the rebuttal, the pros and cons of the HRD reconstruction approaches should be discussed in the related work section. The claim about 120x speedup should be revised and spelled more accurately.